# Setting the Standards: Neonatal Lung Ultrasound in Clinical Practice

**DOI:** 10.3390/diagnostics14131413

**Published:** 2024-07-02

**Authors:** Yogen Singh, Svetlana Dauengauer-Kirliene, Nadya Yousef

**Affiliations:** 1Department of Pediatrics, Division of Neonatology, UC Davis Children’s Hospital, UC Davis Health, Sacramento, CA 95816, USA; 2Department of Human and Medical Genetics, Institute of Biomedical Science, Faculty of Medicine, Vilnius University, Santariškių 2, LT-08661 Vilnius, Lithuania; svetlana.dauengauer-kirliene@mf.vu.lt; 3Division of Pediatrics and Neonatal Critical Care, “A. Béclère” Medical Center, Paris-Saclay University Hospitals, APHP, 92140 Paris, France; nadyayousef.pro@gmail.com

**Keywords:** lung ultrasound (LU), lung ultrasound score (LUS), neonates, ultrasonography

## Abstract

The use of lung ultrasonography in neonates is increasing at a very fast rate. Evidence-based guidelines on the use of lung ultrasound (LU) in neonates and children have been published and well received across the world. However, there remains a lack of standardized curriculum for lung ultrasound training and standards for its application at the bedside. This article focuses on providing a standardized approach to the application of lung ultrasonography in neonates for the common neonatal conditions and how it can be integrated into bedside clinical decision-making.

## 1. Introduction

The use of ultrasound is rapidly increasing in the neonatal intensive care unit (NICU), and now point-of-care ultrasonography (POCUS) is considered an extension of the clinical examination and a “modern stethoscope” [1]. Ultrasonography is safe, non-invasive, and easily available at the bedside, since most modern NICUs today are equipped with ultrasound machines. Thus, POCUS emerges as the ideal first-line investigation to gain anatomical and physiological information in real time to enable physiology-based decision-making at the bedside.

Recently published evidence-based guidelines on the use of POCUS in the neonatal and pediatric intensive care units have provided recommendations for the use of POCUS for assessing five organ systems: heart, lung, brain, abdomen, and vascular access. Of the 45 recommendations, 11 are related to the use of lung ultrasound (LU) in neonates and children [2].

Lung ultrasound (LU) can be safely used to make the diagnosis of the common neonatal respiratory diseases such as respiratory distress syndrome (RDS), transient tachypnea of the newborn (TTN), meconium aspiration syndrome (MAS), pleural effusion, pneumothorax, and lung consolidations, often seen in pneumonia [3,4]. For most respiratory conditions, LU shows superior diagnostic accuracy compared to chest X-rays (e.g., RDS, pneumothorax, consolidation, and pleural effusion) [5,6,7]. LU is also a valuable tool for predicting the need for ventilation and/or surfactant administration, and for monitoring progression towards bronchopulmonary dysplasia (BPD) in preterm infants [8,9]. Neonatologist-performed LU provides accurate real-time information at the bedside, thereby helping the clinician in making the diagnosis accurately and rapidly, and in initiating timely specific therapeutic interventions.

Learning LU is considered simpler and quicker than for some other forms of ultrasound such as echocardiography, and it has been described as having a steep learning curve for most novice users. However, optimal training and experience are still necessary to correctly and consistently diagnose the main pathologies, while minimizing the risk of errors during the interpretation of images and applying the findings in clinical decision-making. Safe and effective application of LU in practice needs a good understanding of technical details and, most importantly, adequate experience in how to integrate the LU findings into the clinical context for providing individualized precision medicine [10,11,12,13].

This narrative review article aims to provide an overview of the practical aspects of performing LU in neonatal care with the intention of helping clinicians to use ultrasound confidently, safely, and effectively for the diagnosis of the most common respiratory conditions in the newborn. It also gives a standardized and simple approach to LU scoring in neonates and provides clear guidance on how LU scores may be used to guide surfactant therapy in the preterm infant.

## 2. Preparing for a Lung Ultrasound Examination

Performing an LU ultrasound examination, as for other procedures in the NICU, requires meticulous preparation and anticipation to optimize the results of evaluation and to minimize any potential negative impact on the patient. Verifying the functionality of the ultrasound equipment and preparing the patient for the procedure are essential considerations to reduce the risk of discomfort or complications. We recommend that a few key steps always be followed to improve the quality of care in neonatal settings.

### 2.1. Communication and Teamwork

Sick neonates are the most vulnerable cohort of the intensive care patient population. Authors advocate for individualized patient-centered care through teamwork with the family and health-care providers in all NICU settings, whenever possible. The purposes and potential benefits of the LU examination should be clearly explained to the parents and team members caring for the patient. Although LU is safe and quick to perform, effective communication and teamwork are still essential for a smooth and efficient process. When possible, the LU examination is performed with the nurse looking after that patient, and parents, when present, to optimize patient comfort and safety.

### 2.2. Preventing Hypothermia

One of the most significant advantages of utilizing LU is that it can be performed without adjustments to the patient’s environment and/or their position. However, the extended duration of any procedure may contribute to changes in body temperature and increase discomfort to patient, especially in sick or extremely preterm neonates. To mitigate the risk of hypothermia, it is imperative to minimize the study time and to activate the incubator’s protective mode, which safeguards the very low birthweight (VLBW) infant from heat loss when the incubator door panels are opened, and to actively use preventive measures against hypothermia. Additionally, after completing the evaluation, any residual ultrasound gel should be gently removed from the skin, and appropriate soft tissues should be used to ensure the infant’s comfort and maintain skin hygiene and integrity. Procedure duration should be as brief as possible to achieve optimal results while minimizing any adverse impacts on the patient.

### 2.3. Patient Positioning

The neonate may be examined in a supine, prone, or lateral position. In the supine patient, the ultrasound examination starts at the anterior chest wall with a vertical longitudinal scan. It is possible to perform a complete lung ultrasound examination with the patient in either the supine or the prone position. It is important to consider the effect of gravity on pathological findings, e.g., free air will be found in the most superior region (anterior of the chest), whilst effusion will be found in the most inferior region. Gravity influences fluid shifts and, consequently, the distribution of artifacts, especially B-lines which reflect the interstitial fluid changes. Generally, waiting approximately for one hour after repositioning the newborn is sufficient for fluid to accumulate in the lowest parts of the lung, thereby allowing for an accurate evaluation. We do not recommend changing the position of infants for the purpose of performing LU and/or for LU scoring. Although most articles on LU scoring in neonates have been reported on LU procedures performed in the supine position, LU scoring can also easily be performed in the prone position to guide the clinical management.

In the case of emergency ultrasound, e.g., in the case of a “crashing” infant, the examination of the anterior chest wall and a quick scan of the posterior-lateral chest wall is sufficient to rule-in or rule-out major pathologies such as pneumothorax and large effusion in a supine patient [14].

### 2.4. Infection Control

Maintaining stringent infection control standards in the NICU is paramount. Both the equipment and the ultrasound operator’s and helpers’ hands must be thoroughly sanitized to prevent the risk of infection. Special wet wipes or non-alcoholic aerosols designed for medical equipment to disinfect the surfaces of the transducers and any leads that are likely to come in contact with the patient, or their immediate surroundings, must be used before and after each ultrasound application. It is vital to select disinfection methods that are safe and effective, without posing any harm to the sensitive neonatal skin or to the ultrasound equipment. Single-use packets of pre-warmed water-based gel should ideally be used for scanning to avoid nosocomial infection risk, but if single-use sachets are unavailable, then correct sealing and storage of gel containers should be ensured according to the manufacturer’s recommendations.

## 3. Practical/Technical Considerations to Performing a Lung Ultrasound Scan

The ultrasound operator should have a good understanding of basic ultrasound physics and how to use the ultrasound machine to acquire optimal quality images.

### 3.1. Choosing the Ultrasound Machine and the Probe

Any ultrasound machine with appropriate probe can be used to perform lung ultrasonography. The interpretation of LU is primarily based upon the analysis of simple artifacts and does not need advanced machine software or expensive US machines with superior image quality. Both a simple handheld and a sophisticated state-of-the-art ultrasound machine can successfully be used to perform lung ultrasonography. LU examinations can be performed with different probes, each with their advantages and their disadvantages [15]. A recent study did not find significant differences in the interpretation of artifacts when using either a hockey stick (linear probe), a microconvex, or a phased-array probe in the newborn [16]. Basic principles of physics should be followed for imaging optimization, such as while obtaining two-dimensional (2D) images, the ultrasound probe should be perpendicular to the tissue of interest. Most clinicians prefer a high-frequency linear or microlinear probe (with a frequency of at least 10–15 MHz) to examine the lungs of the newborn. The biggest advantage of the linear probe is high-quality images of the superficial structures, which allows for a detailed assessment of the pleural line, lung sliding, and microconsolidations (“sub-pleural” consolidations). Microlinear and/or hockey-stick probes also offer the advantage of having a small footprint, and they are easy to use in small preterm infants. Since the high-frequency linear probes offer a superficial vision of the underlying structures, their biggest limitation is lack of power to evaluate deeper structures, and hence difficulty in following lung artifacts deeply, especially B-lines. It may also become more difficult to confidently count individual B-lines when the number of B-lines increases above a certain density [15].

The microconvex probe offers the advantage of having a small footprint and has a good image quality acquisition for both superficial and deep structures in the newborn. It may be considered as a universal probe in the newborn, as it is also used to scan other organs such as the brain, heart, lung, and abdomen. However, more studies are needed to determine its place in neonatal LU compared to the almost universally used linear/microlinear probes. Finally, the phased-array probe offers the advantage of having a small footprint and may be useful when choosing a single probe for holistic ultrasound as it can be used to analyze the heart, head, and abdomen. Although the phased-array probe may be used to recognize the main patterns and LU signs, the image quality is mediocre compared to the linear or microconvex probes, with limited limited visualization of the pleural line which makes the accurate analysis of the pleural line difficult [10]. In adults, the phased array has been to shown to be less accurate than the curved-array probe for lung ultrasonography [17].

### 3.2. Machine Settings and Basic Knobology Useful for Neonatal Lung Ultrasound

Lung ultrasound evaluation is based on the analysis of artifacts, and therefore, machine software and “filters” (e.g., cross-beam, harmonics, etc.) used to decrease image artifacts must be turned off. This is especially important for the evaluation of lung sliding. Many manufacturers have included a “lung preset” in their newer ultrasound machines, but this is not a prerequisite to performing a lung scan. It is possible to use other presets, or to adjust settings in order to obtain a setting where lung sliding and lung artifacts are easy to analyze [10,11].

To obtain high-quality images, there are a few settings that need to be adjusted for an LU scan, primarily the gain, depth, and focal points. The gain should be set to a level where the acoustic shadows of the ribs are black (hypoechoic), the pleural line is white (hyperechoic), and the parietal tissues are gray [18]. There is no recommendation for the optimal depth setting in the newborn, but a depth of 3–4 cm is usually sufficient. When setting the depth, it should be possible to visualize the intercostal space with the pleural line, with usually at least two A-lines, and to be able to follow artifacts sufficiently deeply to be able to distinguish between B-lines and other comet tail artifacts (such as Z-lines, which are considered normal findings on lung ultrasonography). Focal points/focus should be set at the level of the pleural line. When acquiring video clips, the optimal length is between 3 and 6 s.

### 3.3. Ultrasound Modes for Lung Ultrasound

Lung ultrasound is performed using the two-dimensional (2D) mode, also called the B-mode. This is the main mode of imaging for LU. This basic imaging method creates cross-sectional body images through sound waves transmitted and received by a transducer, showcasing their width and depth. Additionally, motion-based mode (M-mode) may be used to evaluate sliding movement of pleura layers during the respiratory cycle (lung sliding). The M-mode provides a mono-dimensional view of the movement of structures at a single line level by displaying real-time motion along the time axis, and it captures details of the movement of structures at a high frame rate (of about 1000 frames per second) [19]. This high-resolution imaging is ideal for assessing fast or subtle movements in tissues and is especially useful when there is a suspicion of pneumothorax, or for the evaluation of pleural effusion [10,11,18].

### 3.4. Machine Setup and Labeling

Completing a thorough, rapid, and efficient lung assessment involves planning, optimal machine setup, and appropriate labeling. Patient information (e.g., patient name and identification number) should be entered before each examination, when time allows it. After adjusting the optimal gain and depth setting, the patient side being examined must be clearly labeled as “Right” or “Left” before the probe is positioned on the corresponding chest side. It is also essential to mark the upper and lower areas being evaluated within the lung. Some new machines provide inbuilt labeling of lung zones which may save time and help in minimizing human errors. Accurate labeling greatly aids in pinpointing the artifacts, making the analysis comprehensive, and it is extremely helpful for reviewing the images later, even outside the clinical setting.

## 4. Performing a Lung Ultrasound Scan Step by Step

### 4.1. Scanning Protocol

In a systematic LU examination, the whole anterior, lateral, and posterior lung regions of each side are scanned [20]. As shown in Figure 1, each lung is divided into three regions:The anterior region, between the sternum and the anterior axillary line.The lateral region, between the anterior and posterior axillary lines.The posterior region, between the posterior axillary line and the spine.

Each region can be further divided into upper and lower parts using the nipple line as a reference. When performing LU scoring, e.g., to evaluate the need for early surfactant, only the anterior and lateral lung regions are scored, as explained below and shown in Figure 1. These lung regions are easy to scan, especially in sick neonates who do not tolerate extensive handling, and they show a good correlation with the severity of interstitial syndrome [21,22]. LU scoring in RDS is described below in detail.

### 4.2. Probe Position

The LU can be performed in two planes (Figure 2):

1. A vertical longitudinal scan (craniocaudal scanning) gives an image of the short axis of the lungs. The probe is placed in a vertical position sagittal to the body axis (Figure 2A). This gives rise to the “Bat sign”, as explained below. This is the most common LU view and considered as the primary mode of scanning for lung evaluation.

2. A horizontal transcostal scan (left–right scanning) gives an image of the lungs in long axis. The probe is placed parallel to the ribs in the intercostal space (Figure 2B). This complementary view avoids the acoustic shadowing of the ribs and can be used to obtain additional information on the extension and/or depth of a pathological lesion or area, e.g., consolidation or areas with interstitial syndrome.

The marker of the ultrasound transducer should be oriented rostrally and match the corresponding side displayed on the screen. This orientation ensures that the upper part of the observed area appears on the left-hand side of the screen, while the lower part is displayed on the right-hand side. In a transcostal/horizontal scan, the right side appears on the left-hand side of the screen, while the left side appears on the right-hand side, as is the convention of most radiological ultrasound examinations (except echocardiography). To avoid any confusion, the right and left sides of the screen are described in relation to the operator’s view. The reverberations occurring at the interface of the pleura and lung are the primary focus of LU. Two surfaces with significantly different acoustic impedances will generate the maximum number of reverberations. Therefore, the transducer should be positioned perpendicular to the lung for optimal image accuracy.

A thorough lung assessment typically begins with systematic sliding technique initiated at the sub-clavicular region, which represents the uppermost part of the lung. The process involves a gradual movement downward to the lower regions, starting from the anterior sections and smoothly transitioning to the lateral areas, and ending at the posterior sections. The evaluation generally extends down to the diaphragm line, marking the abdominal cavity’s boundary. The recognition of this demarcation is important for accurately distinguishing between thoracic and abdominal structures, thereby ensuring a comprehensive lung evaluation [21].

## 5. Principles of Neonatal Lung Ultrasound

Lung ultrasound evaluation is primarily based on the analysis of artifacts. Ultrasound waves cannot transmit through air or bone, and they create artifacts at the interface of tissues with different acoustic properties.

In a normally aerated lung, the ultrasound beams are reflected at the lung surface (interface between the soft fluid-rich tissues of the chest wall and the air-rich lung). This reflection is seen as a hyperechoic line followed by posterior shadowing. The pleural and visceral pleura cannot be distinguished from one another and appear as a single line. Any structure below the pleural line in the normal lung is an artifact [10,11].

The ultrasound waves are stopped by the calcified bones, which produce acoustic shadows beneath the ribs and may hide the underlying artifacts. These rib shadows are landmarks and form the lateral borders of the intercostal space.

Compared to chest X-ray, which provides a global and summative view of lung changes, LU can distinguish between pleural, alveolar, and interstitial lesions.

The main limitations of LU are related to the physical properties of ultrasound. The main limitation of lung ultrasound is that lesions must be in contact with the lung surface to be seen on LU, as ultrasound waves cannot penetrate through the air. In addition, LU cannot be used to evaluate for overdistension in an aerated lung. It is therefore important to remember these limitations and perform a complimentary imaging technique, when necessary, e.g., a chest X-ray if the LU findings do not correspond with the clinical picture or the course.

In practice, most acute lung disease processes, with few exceptions, have a superficial extension to the pleural line making them detectable on LU. Acute lung lesions are usually superficial and extensive. Pleural effusions and pneumothorax always reach the pleural line. Lung consolidations are almost always in contact with the lung surface, and the interstitial syndrome seen at the level of the pleural line is representative of deeper interstitial syndrome in the lungs.

### 5.1. Classical Signs on Lung Ultrasound

When the transducer is placed on the chest wall as previously described, the following underlying structures should be identified on a vertical longitudinal scan on 2D mode:The skin and underlying subcutaneous tissue.The thin layer of thoracic muscle.Ribs, which appear as oval-shaped or arch-shaped echogenic structures accompanied by anechoic vertical shadows. In the newborn, some ribs may not yet be calcified, especially in the anterior chest wall in preterm infants, and therefore no rib shadowing will be seen.Intercostal spaces framed by ribs and marked below by a thin, hyperechoic horizontal pleural line.

### 5.2. The Bat Sign

Finding the “bat sign” is the first step in any systematic LU examination. The “bat sign” is the image formed by an intercostal space consisting of two ribs with acoustic shadowing and the pleural line with its posterior shadowing—first described by Daniel Lichtenstein and it allows for identification of the lung surface [10].

### 5.3. The Pleural Line, Lung Sliding, and the Lung Pulse

The pleural line is visualized as a thin regular and an almost horizontal hyperechoic line, between the two ribs. It is not possible to distinguish between the two pleural leaflets on ultrasound. However, the sliding of the two pleural leaflets against each other is easily detected and is termed as “*lung sliding*”. The evaluation of lung sliding is the most essential part of any LU examination, and the absence of lung sliding is always a sign of underlying pathology that must be explored.

The “*lung pulse*”, which is the transmission of the heart beats, can also often be seen in the newborn, especially in the left anterior lung region, even when the lung is ventilated. The lung pulse resembles lung sliding with a smaller amplitude and a higher frequency. The lung pulse has the same significance as lung sliding in ruling out pneumothorax [23].

A normal pleural line is usually thinner than 1 mm in healthy preterm infants, but measuring the thickness of the pleural line is not usually performed in clinical practice [24,25]. More importantly, accurate measurement of pleural line thickness requires a good image quality since suboptimal scans can give a false impression of a thickened pleural line.

### 5.4. Lung Ultrasound Features and Signs

Lung ultrasound evaluation is primarily based upon the analysis of artifacts, mainly reverberation artifacts or artifacts of repetition, and the real images from lung structures. The main LU signs are described below.

#### 5.4.1. A-Lines

A-lines are the reverberation artifacts. They are reflections of the pleural line, seen as equidistant and static horizontal lines on 2D ultrasound imaging. They reflect the presence of air in the lung.

#### 5.4.2. B-Lines

B-lines are also reverberation artifacts arising at the air/fluid interface of the lung. They provide information on the interstitial fluid component in the lung. B-lines are vertical dynamic well-defined hyperechoic laser-like comet-tail artifacts arising at the pleural line and reaching down to the edge of the screen without fading. They move with lung sliding and erase the A-lines. The presence of more than two B-lines per intercostal space defines an interstitial syndrome.

### 5.5. Consolidations

Consolidations are areas where the lung has lost its aeration (e.g., infection, atelectasis, bleeding, etc.). They are a sign of alveolar involvement and are visible on LU since there is usually some contact with the lung surface. Almost all consolidations have irregular boundaries with the underlying aerated lung called the “shred” sign seen at the deep border of the consolidation (with the exception of translobar consolidations that display the anatomical shape of the lung). Microconsolidations are tiny consolidations found just underneath the pleural line, also called “subpleural” consolidations, that are commonly seen in severe RDS. In many large consolidations, bronchograms may be observed. They are hyperechoic spots, stripes, or areas representing air inside a consolidated lung area. These bronchograms may be dynamic, i.e., show a back-and-forth centrifugal movement synchronous with respiration, or they may be static. It is often difficult to distinguish between static and dynamic bronchograms in the newborn, possibly because of the small tidal volumes during respiration, compared to older patients, and this is rarely a useful criterion in neonatal LU. Atelectasis due to compression, either internal or external, is a form of retractile consolidation, and it is usually not possible to distinguish between atelectasis and other causes of consolidation on lung ultrasonography alone, especially in the newborn infants; the clinical context is essential for making an accurate diagnosis [12,26].

### 5.6. Pleural Effusion

Free fluid in the pleural cavity is easily identified by LU, which has a higher diagnostic accuracy for the diagnosis of effusion than chest X-rays comparable to CT scans. LU can also help in distinguishing between simple and complex effusions. Commonly pleural effusion is found in the lower regions when the probe is positioned in the lung’s lower postero-lateral part in the supine patient, especially at the costophrenic angle for free fluid. To date, there are no major studies regarding the use of LU for effusion in the child or the newborn, and our knowledge derives from adult literature and from personal experience [10,11,24].

In presence of an effusion, the pleural layers are separated by fluid which gives a homogeneous pattern which is usually anechoic or hypoechoic, but it may also be hyperechoic (e.g., for critical causes such as hemothorax or empyema). The “*quad sign*” is useful for the diagnosis of pleural effusion. It is seen as the space bordered by four regular borders which are virtual lines drawn at the level of each rib shadow of the intercostal space, the pleural line (parietal pleura), and the lung line (visceral pleura). The visceral pleura is easily identified when it is separated from the parietal pleura by fluid that allows ultrasound transmission.

Pleural effusion can be categorized on LU as small, moderate, or large in size on visual inspection. Interestingly, an aerated lung floats over the effusion, whilst a consolidated lung floats within pleural effusion since it has approximately the same density.

The inspiratory phase expands the lung, and the spreading of the fluid collection gives a sinusoid appearance on M-mode, which is called the “*sinusoid sign*”. It is especially useful to perform the M-mode scan before deciding whether to aspirate the effusion, since it gives an indication of the space between the pleural layers during the respiratory cycle and hence an idea of whether it is safe or not to insert a needle into the pleural space. In the presence of a very viscous or septate effusion, the sinusoid sign is absent.

## 6. Reporting Lung Ultrasound Findings

Reporting the findings and ultrasonographic impression is an essential part of any ultrasound examination. To avoid errors or missing information, we recommend using a simple systematic and standardized stepwise approach presented in the flowchart in Figure 3. Authors believe that a good practice is to use an “ABCDE” acronym, as described below, and recommend this simple approach for the evaluation, interpretation, and reporting of the LU results. Ideally, the reporting form should detail the essential signs and patterns found on the LU examination, and then propose a conclusion based on LU findings and the clinical situation (Appendix A, Table 1 and Table 2. Reporting is important to document the LU scan for the clinical management, and for peer review. Reporting LU findings may also have medico-legal implications depending on local governance and settings.

Briefly, once the bat sign has been found, the report should start with the results of the pleural line examination including regularity and lung sliding (or lung pulse). Then, the presence of artifacts and signs may be assessed using an ABCDE approach: A = A-lines, B = B-lines (number, arguments for mild/moderate or severe interstitial syndrome depending on combined presence of A and B lines or not), C = consolidation (microconsolidations <0.5 mm or larger consolidations >0.5 mm), D = diaphragm (to ensure that the whole lung has been scanned, and comment on any obvious anomaly of the diaphragm.), E = effusion. Specific LU patterns such as in pneumothorax (see below) should also be reported.

There is currently no universally accepted reporting form for LU findings, but most units use a reporting form, and one such example is shown in Table 1 and Table 2 and in Appendix A. This example has been used in the routine clinical practice for the last few years and has helped in standardizing the reporting in the authors’ institution.

## 7. Applications of LU in the Neonatal Clinical Practice

Lung ultrasound is a simple technique that gives valuable information at the bedside when performed by the clinician. However, all LU findings must be interpreted in the clinical context. Different pathologies may give rise to similar patterns on LU, e.g., severe cardiogenic lung edema in an adult patient may show similar LU findings to severe alveolar–interstitial syndrome seen in the preterm infant with RDS.

This section is focused on recognition of major patterns seen in the common neonatal respiratory conditions. The main LU signs and findings to be reported are shown in Table 3.

(A)Respiratory Distress Syndrome

Respiratory distress syndrome (RDS), previously known as hyaline membrane disease, is one of the most common causes of neonatal respiratory distress and is the result of insufficient surfactant production. It is predominantly, but not exclusively, a disease of the very preterm infants, and its incidence and severity are inversely proportional to gestational age. RDS may coexist with other conditions such as pneumonia, early onset sepsis, or air leaks.

The surfactant deficiency in RDS results in reduced pulmonary compliance and increased alveolar surface tension, leading to alveolar collapse and reduced gas exchange surface area. Characteristic LU findings include preserved lung sliding with an irregular and thickened pleural line, “subpleural” consolidations (or microconsolidations), and numerous « confluent » B-lines, resulting in a bilateral uniform “white lung” appearance (Figure 4). Notably, there are no spared areas, i.e., no areas with a normal pleural line and A-lines, in severe forms of RDS needing surfactant replacement therapy [27].

### Lung Ultrasound Scores to Guide Early Surfactant Administration in RDS

Selective surfactant administration should be performed as soon as possible when indicated or its need is recognized, ideally within the first 2–3 h after birth to improve the outcomes in preterm infants [28]. However, identifying infants within this timeframe may be challenging, since several diagnostic factors such as FiO2 needs may not reach threshold values until a later stage [29]. Lung ultrasound scores, which are based on the change in artifacts reflecting loss of lung aeration, have shown a high predictive value for the need for early surfactant and are therefore increasingly being used to guide surfactant administration.

A recent metanalysis showed pooled estimates of sensitivity of 0.89 (0.82–0.95), specificity of 0.86 (0.78–0.95), diagnostic odds ratio of 3.78 (3.05–4.50), a negative predictive value of 0.92 (0.87–0.97), and a positive predictive value of 0.79 (0.65–0.92) for predicting the first surfactant dose irrespective of the variation in threshold used [30].

The most widely used scoring system is the Lung Ultrasound Score (LUS), first published by Brat et al. [31]. Each lung is divided into three regions (upper and lower anterior, and lateral), for a total of six regions across both lungs (Figure 1). A score between 0 and 3 is given to each region according to the degree of aeration, and a cumulative score is then calculated. A score of 0 is given when there are <3 B-lines per intercostal space, a score of 1 when > or = 3 B-lines per intercostal space, and a score of 2 when there are uncountable (confluent) B-lines with disappearance of A-lines. A score of 3 is given when additionally there is a large consolidation >0.5–1 cm [31]. The LUS is the sum of the regional scores and is inversely correlated with lung aeration. A score >8 in preterm infants below 30 weeks gestational age is highly predictive for the need for surfactant therapy [26]. LUS-guided surfactant replacement therapy (also known as (Echography-guided Surfactant THERapy (ESTHER)) may significantly increase the number of patients treated within the optimal time frame of 3 h after birth, reduce oxygen exposure early in life, and improve oxygenation after surfactant dosing, without increasing the use of surfactant or changing cost/benefit ratios [31,32].

Another method of scoring is the image classification system based on three lung ultrasound profiles (LP) with the following characteristics: Type 1 for a “white lung” image with uncountable “confluent” B lines); Type 2 for a mixed pattern with A and B lines; Type 3 for normal lung imaging (A lines and pleural sliding present). This scoring system is accurate in predicting the need for the first dose of surfactant with an AUC of 0.88 and an optimal specificity and sensitivity of 0.86 [33].

Both the LUS and the image classification system show good accuracy for predicting the first dose of surfactant in preterm infants, although most published studies have used the LUS or similar scoring systems.

(B)Meconium Aspiration Syndrome (MAS)

Meconium aspiration syndrome (MAS) is an important cause of neonatal respiratory distress and carries a high burden of mortality and morbidity worldwide. Inhalation of meconium-stained amniotic fluid results in obstruction and collapse of the airways, while its chemical properties may lead to inflammation, surfactant inactivation, and alveolar injury [34]. Severe cases of MAS usually lead to significant lung injury and progression towards neonatal ARDS [35].

Classic lung ultrasound findings in MAS show a heterogenous distribution of lung lesions with consolidations of variable sizes, areas with interstitial syndrome, and mild to moderate pleural effusion [36,37]. Imaging of the same patient may reveal normal lung areas, interstitial patterns, and areas of consolidation and atelectasis, as shown in Figure 5. These findings are dynamic and reflect the underlying disease process. The management of severe MAS may be challenging. Serial scanning offers invaluable information regarding response to management and changes in lung aeration, which may be helpful in guiding respiratory management and screening for complications such as pneumothorax.

(C)Transient Tachypnea of Newborn (TTN)

Transient tachypnea of the newborn (TTN) is a common cause of respiratory distress in the newborn period, commonly seen soon after birth, that often presents in near-term or term infants, especially if born after cesarean section and/or in infants of diabetic mothers [38]. The lung edema observed in TTN is the result of delayed alveolar fluid resorption and clearance during the transitional period, and LU findings in TTN correspond well with the underlying physiopathology. Classic TTN findings on LU were first described by Copetti et al. and show areas with multiple B-lines, i.e., interstitial syndrome in the lower lung fields, and normal or near-normal upper lung fields [39]. A sharp transition between the two patterns (area with interstitial syndrome with B-lines meeting and spared or normal area with A-lines, both with preserved lung sliding, called the “double lung point” is pathognomonic of TTN when present. The “double lung point” may be seen in approximately 50% of cases and sometimes appears later in the course of the disease (Figure 6). Contrary to RDS, where there are no spared areas, i.e., no areas with normal or almost normal aeration, TTN always shows some spared areas with A-lines, a thin and well-defined pleural line, and no consolidations [40]. An LU aeration score shows good correlation with increased work of breathing in TTN and can be used to monitor the clinical course [40].

(D)Pneumothorax

Lung ultrasound has an excellent diagnostic accuracy in diagnosing pneumothorax, especially in the newborn, and has a higher sensitivity and specificity than chest X-ray in recognition of pneumothorax. A recent meta-analysis by Dahamarde et al. reported a sensitivity of 96.7% and a specificity of 100% for the neonatal population, which is even higher than adults, where the sensitivity is 82.9% and the specificity is 98.2% [41]. Current guidelines recommend using LU for both diagnosing pneumothorax in children and neonates and to guide thoracentesis [42].

Like adults and older children, the three cardinal signs of pneumothorax are: (1) absence of lung sliding (or lung pulse), (2) absence of B-lines (or consolidation), and (3) “Stratosphere” or Barcode” sign on M-mode [43] (Figure 7).

The lung point, when present, is specific to pneumothorax [44]. It is almost a longitudinal line seen at the transition between the pneumothorax and the normal lung with no pneumothorax. Two simultaneous patterns are seen on LU, separated by the lung point: (1) one with abolished lung sliding and only A-lines (on the side of pneumothorax), and (2) the other with lung sliding (over the “normal” lung with no pneumothorax).

Making the diagnosis of pneumothorax requires a stepwise approach. Air collects in the most superior area, near the anterior chest wall in the supine patient. To diagnose pneumothorax, the ultrasound probe is first placed on the anterior chest wall, and the lung surface is identified (using the “bat sign”). Then, the pleural line is evaluated for the presence of lung sliding. If lung sliding is abolished, and the only visible artifacts are the A-lines, then there is a high suspicion for pneumothorax. The M-mode shows a classical “stratosphere sign” or “barcode” sign, which is different from the normal “seashore” sign. The stratosphere sign shows only horizontal hypoechoic and hyperechoic parallel lines, with no difference seen between the structures superficial and deep to the pleural line.

Once absence of lung sliding has been confirmed, the next step is to search for the lung point, which confirms the diagnosis of pneumothorax with 100% specificity, but which may be absent in a complete or large pneumothorax when the lung does not reach the chest wall and therefore is not detected by ultrasound [45]. The position of the lung point also gives an estimation of the size of the pneumothorax—small (lung point anterior to the anterior axillary line), moderate (lung point between the anterior and posterior axillary line), and large (lung point posterior to the posterior axillary line). The presence of lung sliding, lung pulse, B-lines, or consolidation rules out pneumothorax.

## 8. Conclusions

This review article on lung ultrasonography provides an overview of the essential technical skills, imaging acquisition, artifacts, and a suggested standardized approach to interpretation. An overview of the LU findings in the common neonatal respiratory conditions has been provided. LU is rapidly being adopted in neonatal practice and is transforming the management of preterm and term infants with respiratory distress. It is a quick and safe imaging technique that offers many advantages over chest X-rays, but it should always be interpreted in the clinical context. A standardized and systematic approach to image acquisition, interpretation, and reporting are essential to optimize the quality of the examination and to minimize errors.

## Figures and Tables

**Figure 1 diagnostics-14-01413-f001:**
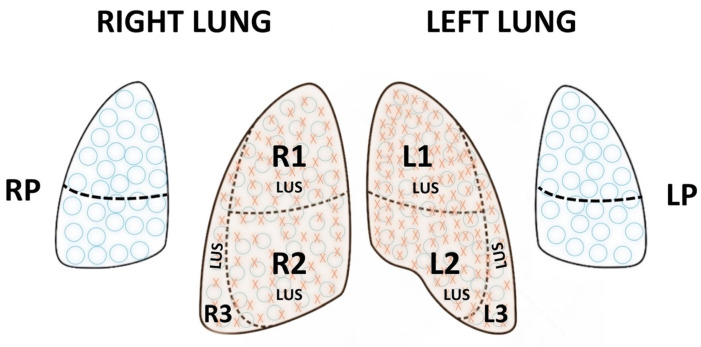
Lung regions for a systematic lung ultrasound (LU) examination and for lung ultrasound scoring (LUS). Schematic illustration of lung regions in a full LU in neonates represented by blue circles 
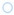
. R1 is the right upper anterior region, R2 is the right lower anterior region, R3 is right lateral, L1 is the left upper anterior region, L2 is the left lower anterior region, and L3 is left lateral, whilst RP and LP are the right and left posterior areas, respectively. The areas used for LUS are represented by red crosses in addition to the blue circles 
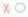
 and comprise R1, R2, R3 and L1, L2, and L3.

**Figure 2 diagnostics-14-01413-f002:**
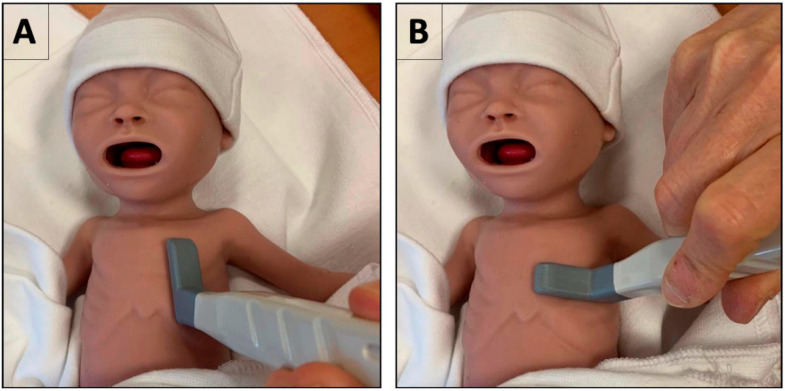
Probe position for performing lung ultrasound (LU)—Illustration on how to position the ultrasound probe for performing LU. Panel (**A**) shows a vertical longitudinal scan, most commonly used in neonates, and Panel (**B**) shows a horizontal transcostal scan.

**Figure 3 diagnostics-14-01413-f003:**
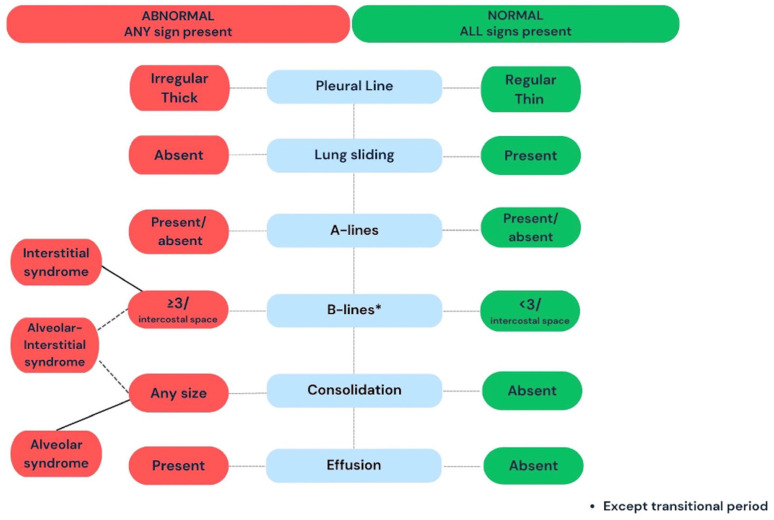
A schematic illustration of the systematic approach to the basic steps involved in lung evaluation. Lung ultrasound scanning follows a systematic sequential order. The examination starts with the identification of the lung surface (the Bat sign), then the evaluation of the pleural line, and the presence of lung sliding. Next, the presence of artifacts and signs are evaluated. A lung scan is normal when the pleural line is regular and thin with lung sliding, less than 3 B-lines per intercostal space, no consolidation, and no pleural effusion. A thick irregular pleural line, absent lung sliding, 3 or more B-lines per intercostal space (transitional period excepted), and consolidation are all pathological findings and warrant further workup or follow-up depending on the clinical context.

**Figure 4 diagnostics-14-01413-f004:**
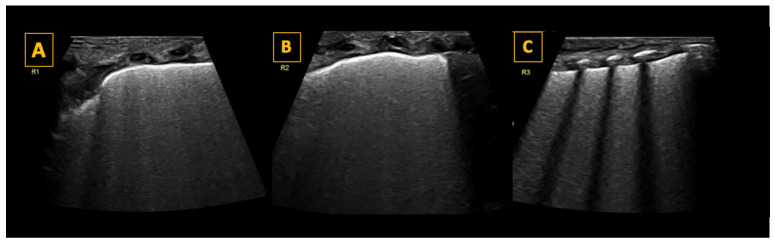
Lung ultrasound findings in respiratory distress syndrome (RDS). Panel (**A**–**C**) show longitudinal scans of the right anterior superior (R1), the right anterior inferior (R2), and the right lateral lung regions, respectively, of a preterm infant with RDS. As shown in the panels, the pleural line is thickened, and all lung fields show a universal “white lung” appearance with no visible A-lines reflecting severe loss of aeration in RDS. LUS scoring reveals a score of 2 in each lung field, with a total score of 6 for the right lung. In this patient, LUS was also 6 for the left lung, giving a total LUS score of 12/18.

**Figure 5 diagnostics-14-01413-f005:**
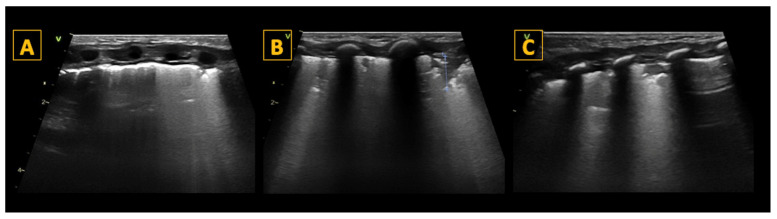
Meconium aspiration syndrome (MAS). Panel (**A**–**C**) (longitudinal scan) show typical LU findings in a neonate presenting with MAS. LU scanning reveals heterogeneous findings related to injury and obstruction by meconium. Panel (**A**) shows an irregular pleural line with small consolidations and multiple B-lines. Panel (**B**) shows multiple consolidations of variable sizes with an otherwise white-looking lung. The largest consolidation is shown with the blue dotted line. Panel (**C**) shows one intercostal space with a normal-looking appearance adjacent to a lung area with multiple B-lines and a consolidation reflecting the non-homogenous nature of the lung injury.

**Figure 6 diagnostics-14-01413-f006:**
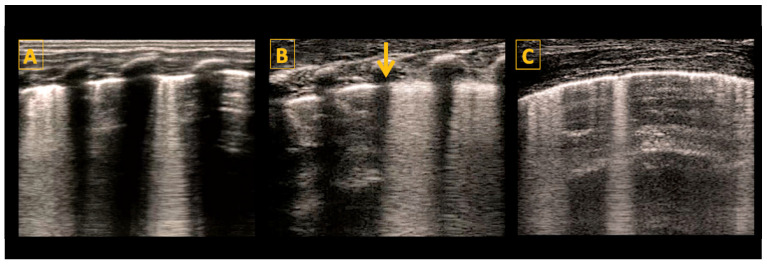
Lung ultrasound findings in transient tachypnea of the newborn (TTN). The lung shows signs of incomplete fluid resorption with a mix of A- and B-lines and a well-defined and thin pleural line in aerated areas. Panel (**A**) is a longitudinal scan showing lower intercostal spaces with B-lines (interstitial fluid) alternating with upper intercostal spaces with A lines (well-aerated lung areas) reflecting incomplete and non-homogenous lung fluid resorption. Panel (**B**) shows a sharp distinction between well-aerated areas and areas with B-lines called the “double lung point” (yellow arrow), which is seen in approximately 50% of TTN cases. Panel (**C**) shows a horizontal scan showing the typical mix of A- and B-lines in TTN with a sharp well-defined pleural line.

**Figure 7 diagnostics-14-01413-f007:**
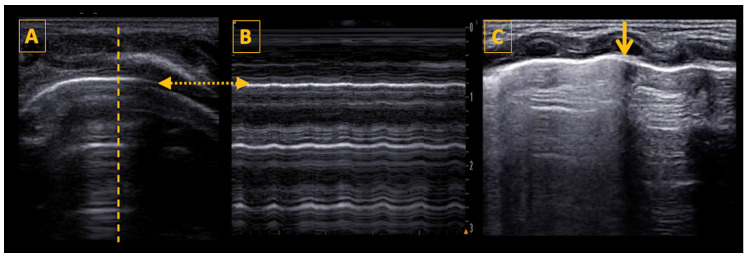
Lung ultrasound findings in pneumothorax. Panel (**A**) shows a typical image of neonatal pneumothorax with a thin normal pleural line, A-lines, and no B-lines. The thatched line shows the position of cursor for the M-mode scan. Panel B shows the M-mode scan revealing the stratosphere (or barcode) sign, a pattern confirming loss of lung sliding. The thatched horizontal arrow shows the position of the pleural line in both panels (**A**,**B**), which is necessary to determine for correct image interpretation. Panel (**C**) shows a longitudinal scan with a double pattern on both sides of the lung point, which is marked by the vertical yellow arrow. On the right side of the arrow (left side of the patient), there are only A-lines visible, on the left side of the arrow the lung appears fuzzy because of the presence of lung sliding and B-lines. The lung point has a specificity of 100% for the diagnosis of pneumothorax.

**Table 1 diagnostics-14-01413-t001:** An example for a systematic report on lung ultrasound findings.

	Regular Pleural Line	Lung Sliding	A Lines	B Lines *			Consolidations		Effusion	Ptx	LUS
				<3	≥3	Confluent	<5 mm	≥5 mm			
R1											
R2											
R3											
RP											
DIAPHRAGM:											
NOTES:											
CONCLUSION: Lung aeration (LUS)— Pathophysiologic syndrome—											
	**Regular** **Pleural Line**	**Lung Sliding**	**A lines**	**B Lines ***			**Consolidations**		**Effusion**	**Ptx**	**LUS**
				**<3**	**≥3**	**Confluent**	**<5 mm**	**≥5 mm**			
L1											
L2											
L3											
LP											
DIAPHRAGM:											
NOTES:											
CONCLUSION: Lung aeration (LUS)— Pathophysiologic syndrome—											
MAIN CONCLUSION AND SUGGESTED DIAGNOSIS:											

*—number of B lines are per one intercostal space. Ptx—pneumothorax. R1—right upper anterior area, R2—right lower anterior area, R3—right lateral area, RP—right posterior area, L1—left upper anterior area, L2—left lower anterior area, LP—left posterior area. LUS—lung ultrasound score. Grey cells indicate not applicable.

**Table 2 diagnostics-14-01413-t002:** An example for a systematic report showing findings of lung ultrasound and lung ultrasound scoring.

	Regular Pleural Line	Lung Sliding	A Lines	B Lines *			Consolidations		Effusion	Ptx	LUS
				<3	≥3	Confluent	<5 mm	≥5 mm			
R1	-	+	-	-	+	+	+	-	-	-	2
R2	-	+	-	-	+	+	+	-	-	-	2
R3	-	+	-	-	+	+	+	-	-	-	2
RP	-	+	-	-	+	+	+	+	-	-	
Diaphragm: symmetrical movements, normal appearance.											
Notes: nonvisible pleural line in R3, RP.											
CONCLUSION: Lung aeration (LUS)—“white” lung appearance, poor aeration in R3, RP LUS 6. Pathophysiologic syndrome—homogenous alveolo—interstitial syndrome.											
	**Regular** **Pleural Line**	**Lung Sliding**	**A lines**	**B lines ***			**Consolidations**		**Effusion**	**Ptx**	**LUS**
				**<3**	**≥3**	**Confluent**	**<5 mm**	**≥5 mm**			
L1	-	+	-	-	+	+	+	-	-	-	2
L2	-	+	-	-	+	+	+	+	-	-	3
L3	-	+	-	-	+	+	+	+	-	-	3
LP	-	+	-	-	+	+	+	+	-	-	
Diaphragm: symmetrical movements, normal appearance.											
Notes: non visible pleural line L3, LP;											
CONCLUSION: Lung aeration (LUS)—“white” lung appearance, poor aeration in L3, LP. LUS 8 Pathophysiologic syndrome –homogenous alveolo—interstitial syndrome.											
MAIN CONCLUSION AND SUGGESTED DIAGNOSIS: Homogenous “white” lung. Alveolo-interstitial syndrome with poor aeration in lateral and posterior parts, presenting patterns suggestive of severe RDS. Total LUS of 14 suggestive for surfactant need. Clinical correlation is essential for an accurate diagnosis.											

*—number of B lines are per one intercostal space. Ptx—pneumothorax. R1—right upper anterior area, R2—right lower anterior area, R3 -right lateral area, RP—right posterior area, L1—left upper anterior area, L2—left lower anterior area, LP—left posterior area. LUS—lung ultrasound score. Grey cells indicate not applicable.

**Table 3 diagnostics-14-01413-t003:** Main lung ultrasound findings in neonatal respiratory diseases.

Lung Pathology		Lung Sliding	A-Lines	B-Lines	Consolidation		M-Mode	Other Signs
					Microconsolidations <5 mm	>5 mm		
Normal lung	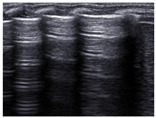	Present	Yes	<3/intercostal space	No	No	Seashore sign	Lung pulse may be present
RDS	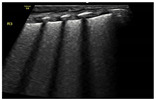	Present	No (minimal)	>3/intercostal space Confluent “White lung”	Usual	Unusual	Seashore sign	No “spared” areas
TTN	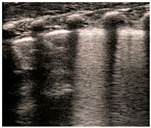	Yes	Yes	Yes	Sometimes	Unusual	Seashore sign	“Spared areas” Double lung point in approximately 50% of cases
Pneumothorax	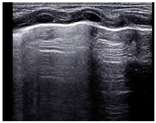	Absent	Yes	No	No	No	Stratosphere sign	The lung point confirms diagnosis
Meconium aspiration syndrome	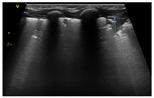	Variable	Variable	Yes	Yes/no	Yes	Seashore sign	Multiple consolidations usual

## Data Availability

No new data were created or analyzed in this study. Data sharing is not applicable to this article.

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
