# Peer review of "Setting the Standards: Neonatal Lung Ultrasound in Clinical Practice"

_diagnostics, 2024, doi:10.3390/diagnostics14131413_

Round 1

Reviewer 1 Report

Comments and Suggestions for Authors

The authors reviewed the lung sonography in neonates and children. Standarized curriculum and standards for bedside application from their practice experiences are summarized and shared. This review is quite useful for the clinical sonographer.

How many hospitals or medical centers are involved? Are the standards summarized in this study widely accepted by the peers, especially those in different countries?

Are there any challenges that cannot be solved using lung sonography?

All figures are not included in the manuscript.

Remove some additional and unnecessary spaces in the text. While a space is required  between the citation and the word in sentences.

Very a few sentences are incomplete.

Author Response

REVIEWER 1

The authors reviewed the lung sonography in neonates and children. Standarized curriculum and standards for bedside application from their practice experiences are summarized and shared. This review is quite useful for the clinical sonographer.

Response: We would like to thank the reviewer for the encouraging feedback.

Question: How many hospitals or medical centers are involved? Are the standards summarized in this study widely accepted by the peers, especially those in different countries?

Response: Thank you for your important question. We realize that this information may not be clear from the manuscript.

The authors come from three different medical centers, in three different countries with variations in clinical practice. Two of the authors are co-authors of the international evidence-based guidelines for point of care ultrasound in the critically ill neonates and children published by the European Society of Pediatric and Neonatal Intensive Care. (Singh 2020) and they have published extensively on POCUS and lung ultrasound in neonates.

The standards proposed here are widely accepted by peers internationally and represent the current standard of performing neonatal lung ultrasound in many units where experts in lung ultrasound work.

The reporting form shown in tables 2 and 3, and annex 1 are an example used in one of the centers. They are meant as an example of a systematic reporting form. We have added a short paragraph to explain the lack of universal reporting form and that this an example use in one of the centers. (lines 497-499)

Question: Are there any challenges that cannot be solved using lung sonography?

Response: Thank you for pointing out that we have not addressed the limitations of using lung ultrasound, which is essential. We have added a paragraph on the limitations of using lung ultrasound (lines 294-300).

Question: All figures are not included in the manuscript.

Response: Please accept our apologies for this oversight. All figures are now included within the manuscript.

Question: Remove some additional and unnecessary spaces in the text. While a space is required  between the citation and the word in sentences. Very a few sentences are incomplete.

Response: We thank the reviewer for pointing out the editing mistakes that we missed or overlooked. Apologies! We have gone through the manuscript carefully and corrected any superfluous spaces and incomplete sentences. We hope not to have left any uncorrected mistakes.

Reviewer 2 Report

Comments and Suggestions for Authors

This is a well written review by an international group of authors. It provides a guideline of structured approach to lung ultrasound in neonates.

A minor point: Heading 2.2. Evaluation of patient's immediate environment. 

This heading is too broad for what was described, i.e. prevention of hypothermia and cleaning of gel. Readers may conjure up wide range of variables like noise, light, humidity etc. associated Immediate environment. Maybe a simple term of "Prevention of hypothermia" would suffice. 

Author Response

REVIEWER 2

This is a well written review by an international group of authors. It provides a guideline of structured approach to lung ultrasound in neonates.

Response: We would like to thank the reviewer for their positive feedback.

Comment: A minor point: Heading 2.2. Evaluation of patient's immediate environment. 

This heading is too broad for what was described, i.e. prevention of hypothermia and cleaning of gel. Readers may conjure up wide range of variables like noise, light, humidity etc. associated Immediate environment. Maybe a simple term of "Prevention of hypothermia" would suffice. 

Response: We agree with the reviewer. The title has now been changed to “Prevention of hypothermia” as suggested by the reviewer. Thank you.